

# Global RNA editome landscape discovers reduced RNA editing in glioma: loss of editing of gamma-amino butyric acid receptor alpha subunit 3 (GABRA3) favors glioma migration and invasion

Vikas Patil[1,2], Jagriti Pal[1], Kulandaivelu Mahalingam[2] and Kumaravel Somasundaram[1]

[1] Department of Microbiology and Cell Biology, Indian Institute of Science, Bangalore, India
[2] Department of Bio-Medical Sciences, School of Biosciences and Technology, Vellore Institute of Technology, Vellore, India

Corresponding author
Kumaravel Somasundaram,
skumar1@iisc.ac.in

## ABSTRACT

**Background:** Gliomas are the most common and lethal type of intracranial tumors. With the current treatment regime, the median survival of patients with grade IV glioma (glioblastoma/GBM) remains at 14–16 months. RNA editing modifies the function and regulation of transcripts. The development of glial tumors may be caused by altered RNA editing events.

**Methods:** In this study, we uncover the global RNA editome landscape of glioma patients from RNA-seq data of control, lower grade glioma (LGG) and GBM samples ($n = 1{,}083$).

**Results:** A-to-I editing events were found to comprise 80% of the total editing events of which 96% were located in the Alu regions. The total RNA editing events were found to be reduced in glioma compared to control samples. More specifically, we found Gamma-aminobutyric acid type A receptor alpha3 (GABRA3) to be edited (c.1026 A-to-G; pI343M) in 73% (editing ratio 0.8) of control samples compared to LGG (28.96%; 0.47) and GBM (5.2%; 0.53) samples. GABRA3 transcript level was found to be downregulated in glioma compared to control in a grade-specific manner with GBMs having the lowest level of the transcript. Further, GABRA3 transcripts were observed to be higher in edited compared to unedited glioma samples. The transcript and protein levels of exogenously expressed gene were found to be higher for edited compared to unedited GABRA3 in glioma cells. Further, exogenously expressed edited GABRA3 inhibited migration and invasion of glioma cells efficiently but not the unedited GABRA3.

**Conclusion:** Collectively, our study discovered a reduction in RNA editing during glioma development. We further demonstrate that elevated RNA editing maintains a high level of GABRA3 RNA and protein in normal glial cells which provides a less migratory environment for the normal functioning of the brain. In contrast, the reduction in GABRA3 protein levels, due to lower stability of unedited RNA, results in the loss of function which confers an aggressive phenotype to GBM tumor.

# INTRODUCTION

Thirty percent of all central nervous system tumors and eighty percent of malignant brain tumors are composed of glial cells and are called gliomas (*Goodenberger & Jenkins, 2012*). Astrocytomas are the most common and lethal type of glioma. They are divided into four categories as per WHO classification based on histopathology that is, grade I, II, III and IV. Grade I, also known as, Pilocytic astrocytoma is benign, while grades II to IV are progressively more malignant. The most aggressive grade IV is called Glioblastoma (GBM). The median survival of GBM achieved till today through surgery, chemotherapy and radiotherapy is only 14.6 months (*Stupp et al., 2009*). Hence, further studies to understand the molecular pathways deregulated in GBM is important.

RNA editing is a molecular process by which RNA sequences are altered post-transcriptionally through base conversion or insertion/deletion. RNA editing increases proteomic diversity in cancer. In mammals, especially in humans, the most common type of editing changes include A-to-I and C-to-U base modifications (*Gott & Emeson, 2000*). A-to-I occur in a large number of transcripts including miRNA and it is carried out by ADAR family of enzymes (*Keegan, Gallo & O'Connell, 2001*; *Nishikura, 2010*; *Wulff, Sakurai & Nishikura, 2011*). C-to-U editing events occur relatively rarely and are carried out by a particular deaminase family of enzymes called APOBEC (*Rosenberg et al., 2011*). ADAR enzyme binds to double-stranded RNAs and deaminates adenosine to inosine. Inosine in turn is recognized as guanosine by the cellular machinery. A-to-I editing events are most common in Alu repeats because of the double stranded RNA structures formed by inverted Alu repeats that spread across the genome (*Levanon et al., 2004*; *O'Connell et al., 1995*). RNA editing can lead to various changes in the mRNA including amino acid change, alteration in splice site, RNA stability, changes in secondary structures leading to alterations in proteins binding to it, alteration in miRNA binding etc.

In the present study, we propose to understand the role of RNA editing in glioma development. Until the last 6 years, various groups have identified RNA editing events by protein sequencing or cDNA sequencing of individual genes. Reduced A-to-I editing was observed in several human tumor types, including brain tumors (*Maas et al., 2001*; *Paz et al., 2007*). Also, it was observed that restoration of the defective editing activity was able to inhibit proliferation of brain tumor cells (*Cenci et al., 2008*; *Paz et al., 2007*). Large scale mRNA sequencing was carried out to determine the regulation of RNA editing during brain development (*Wahlstedt et al., 2009*). Subsequently, the advent of high-throughput sequencing technology led to the study of RNA editome landscape in different diseases including cancer. For example, increased A-to-I editing in hepatocellular carcinoma (HCC) has been identified in recent studies (*Chan et al., 2014*; *Chen et al., 2013*). Increased editing of AZIN1 transcript resulting in amino acid substitution was observed in HCC and the above change was seen to confer enhanced tumorigenicity (*Chen et al., 2013*). RNA sequencing and bioinformatic analysis have facilitated identification of

novel RNA editing sites (*Bahn et al., 2012*; *Choudhury et al., 2012*; *Ju et al., 2011*; *Li et al., 2011*).

The importance of RNA editing in glioma pathogenesis was reported in an earlier study where authors studied the role of ADAR2 in GBM growth (*Galeano et al., 2013*). In a more recent study, *Silvestris et al. (2019)* used the inosinome profile for patient stratification in GBM. In this study, we have carried out comprehensive analysis of RNA sequencing data of normal brain, lower grade glioma (LGG) and GBM samples from publicly available datasets. The data was carefully analyzed to understand the distribution of RNA editing events across different regions of the genome in normal and glioma samples. Comparison of editing events between normal, LGG and GBM samples was performed to evaluate the global regulation patterns of RNA editing in normal vs. diseased conditions. Regulation of ADAR enzymes in gliomas and the differential editing events between the different types of gliomas was queried to understand how the editing enzymes are regulated and how the regulation correlates with editing during glioma progression. Pathway analysis of differentially edited genes was performed to find out cellular processes regulated by editing in glioma. This revealed important genes involved in brain functions. Under-editing of glutamate ionotropic receptor AMPA type subunit 2 (GRIA2) has been reported to play a role in glioma aggressive phenotype (*Maas et al., 2001*; *Oakes et al., 2017*). Here, we further studied the effect of missense RNA editing in gamma-amino butyric acid receptor alpha subunit 3 (GABRA3), a gene involved in neuronal signaling in the brain. Experimental investigation revealed that loss of editing in this gene in glioma leads to a more aggressive tumor phenotype.

## MATERIALS AND METHODS

### Data acquisition

We downloaded (downloaded in March 2014) 529 lower-grade glioma and 172 GBM RNA and exome sequencing samples from TCGA through the Cancer Genomics Hub (CGHub, dbGaP accession number phs000178). RNA sequencing data of 45 glioma cell lines were also downloaded from CCLE through CGHub (dbGaP accession number phs000178). Samples downloaded from TCGA and CCLE study were in BAM format, we converted BAM format to fastq format and used for further analysis. We also downloaded 174 LGG and 100 GBM RNA sequencing samples from Chinese Genome Glioma Atlas (CGGA) via SRA (SRP027383) (*Bao et al., 2014*). RNA sequencing data of control brain samples (SRP033725, SRP045638, SRP044668) ($n = 63$) were downloaded from SRA (*Akula et al., 2014*; *Gill et al., 2014*; *Jaffe et al., 2015*). Data downloaded from SRA was in sra format; we converted sra format to fastq format using SRA Toolkit. Sample information is provided in Table S1.

### RNA editing pipeline

BWA aligner (*Li & Durbin, 2009*) was used to align the RNA-seq reads on hg19 and Ensemble164. The duplicate removal was carried out using picard (*Picard toolkit, 2014*, http://broadinstitute.github.io/picard/) followed by re-alignment and base re-calibration. GATK unifiedgenotyper tool (*McKenna et al., 2010*) was used in order to call the editing

events from RNA-seq reads which was compared to hg19. Next, the total variants obtained were filtered and also the potential SNPs were removed by dbSNP (*Sherry et al., 2001*), 1,000 g (*Genomes Project et al., 2015*) and ESP6500 databases (*NHLBI, 2014*, http://evs.gs.washington.edu/EVS). For each read first six bases were discarded so as to remove the mismatches which were caused due to random-hexamer priming. Following steps were used to filter out the spurious changes: (a) Editing events were removed which showed changes within the 4 bps of known splice junction. (b) Editing events present in homopolymer runs of ≥5 were removed. (c) Editing events present in different locations with high similarity found in BLAT (*Kent, 2002*) were also removed.

## Editing ratio calculation

Coverage is very much important factor to predict accurate RNA editing events. If variant position was having 10 or more reads then only that position was considered for calculating editing ratio. Editing ratio was calculated by dividing alternate allelic depth by total depth for that base. If a position was having editing ratio greater than 0.2 then that position was considered as putative RNA editing events.

## Differential RNA editing

We compared RNA editing ratios between control brain samples and tumor samples to find out differentially edited events. We calculated average editing ratios for both the conditions. Editing difference was calculated by subtracting average value of control brain samples from average value of tumor samples. Significant testing was carried out by using Mann–Whitney $U$-test. If RNA editing event follow the criteria—(a) Benjamini/Hochberg FDR correction value is less than 0.05, (b) absolute difference between average RNA editing of tumor samples vs. control brain samples is greater than 0.2 and (c) absolute difference in percentage of samples edited, tumor vs. control brain, is more than 10% then that RNA editing event was considered as significantly differential RNA editing event.

## Pathway analysis

Significant differentially edited events (glioma vs. control brain sample) were used for gene ontology (GO) analysis using DAVID bioinformatics resources 6.7 (Database for Annotation, Visualization and Integrated Discovery) (*Da Huang, Sherman & Lempicki, 2009*). Missense editing events were identified as editing events that cause a change is protein sequence of the gene.

## Cell lines, constructs and antibodies

Glioma cell lines were obtained from ECACC, Salisbury, UK. All cells were cultured in Dulbecco's Modified Eagles' Medium (DMEM) containing 10% Fetal Bovine Serum (FBS). The cells were grown at 37 °C in 5% CO2.

Antibodies used include anti-GABRA3 (HPA000839-100UL; Sigma–Aldrich, St. Louis, MO, USA), DDK (#TA-50011; Origene, Rockville, MD, USA), Actin (#A3854; Sigma–Aldrich, St. Louis, MO, USA). GABRA3 overexpression construct: (NM_000808) Human cDNA ORF Clone with catalog number RC206286.

## Site directed mutagenesis

pCMV-Entry-GABRA3 plasmid was subjected to SDM using QuikChange Multi Site Directed Mutagenesis Kit (Catalog no. 200515). A total of 100 ng of plasmid was taken and mixed with buffer, dNTPs, mutation primer and Pfu polymerase enzyme mix. During PCR reaction, extension was performed at 65 °C for 14 mins (2 mins/kb of plasmid length). The whole mix was then incubated with DpnI enzyme at 37 °C for 1 h. A total of 2 µl of the reaction mix was transformed in *Escherichia coli* DH5α and plated on Kanamycin LB agar plate. Mutant colonies were picked and plasmid was isolated. The mutation status was verified by Sanger sequencing.

## Transfection of plasmid DNA in glioma cell lines

A total of $0.5 \times 10^6$ cells were plated in 35 mm dishes. A total of seven µg of each of VC or GABRA3 plasmids were transfected using Lipofectamine 2000™ in OptiMEM™ medium. After 6 h of transfection, the medium was changed to DMEM containing 10% FBS. Cells were plated for experiment or harvested after 24 h of transfection for RNA or protein isolation.

## Boyden-chamber assay for cell migration

For every experimental condition, 50,000 cells were plated in triplicates in the upper-chamber (Boyden chamber from BD Biosciences) in serum-free medium. In the lower chamber, DMEM containing 10% FBS was added. The cells were allowed to migrate for 6–8 h. Next, cells were fixed in 100% chilled methanol and stained using 0.2% crystal violet. The experiment was carried out in two biological replicates and the quantification is shown for one experiment. Unpaired *t*-test was used for calculating statistical significance between two groups. *, ** and *** denote *p* value of <0.5, <0.1 and <0.001 respectively.

## Boyden-chamber matrigel assay for cell invasion

Activation of the matrigel chambers (BD Biosciences, Franklin Lakes, NJ, USA) were performed by incomplete DMEM for half an hour. For each experimental condition, 75,000 cells were plated in the upper chamber in serum-free medium. A total of 10% FBS was added in the lower chamber. The cells were allowed to invade for 20–22 h after which cells were fixed and stained using 0.2% crystal violet stain. The experiment was carried out in two biological replicates and the quantification is shown for one experiment. Unpaired *t*-test was used for calculating statistical significance between two groups. *, ** and *** denote *p* value of <0.5, <0.1 and <0.001 respectively.

## RNA isolation, cDNA conversion and real-time qPCR

Cells were harvested using Trizol™ and RNA was isolated immediately following phenol-chloroform extraction method. The RNA was quantified using nanodrop method and quality assessment was performed by running 1 µg of RNA on agarose-formaldehyde gel. A total of 2 µg good quality RNA was used per reaction for cDNA conversion. Applied Biosystems™ High Capacity cDNA Reverse Transcription kit (Part no. 4368813) was used. The cDNA strand synthesis was carried out in Biorad S1000™ Thermal Cycler.

Thermo Scientific's DyNAmo reagent was used for real time qPCR. Applied Biosystems™ 7900HT Fast Real-Time PCR system was used to measure RNA levels. GAPDH was used as internal control. The experiment was carried out in two biological replicates and the quantification is shown for one experiment.

The primer sequences used in the current study:
CMV promoter FP: GCTCGTTTAGTGAACCGTCAG
GABRA3 5′ UTR FP: CAGTCACACCACAGCGTCT
GABRA3 Common RP: GGTTCCAGGGAGAATATTAATCAGG
GAPDH FP: GTCTCCTCTGACTTCAACAGCG
GAPDH RP: ACCACCCTGTTGCTGTAGCCAA

## Protein isolation and western blotting

Cells were trypsinized and pelleted down at 4 °C. RIPA lysis buffer containing 1 mM NaF, 1 mM PMSF and 1X SIGMA*FAST*™ protease inhibitor cocktail was used to lyse the cells. After 1 h, the cell debris were pelleted at 14,000 rpm and 4 °C for 30 min. The protein concentration of each sample was determined by Bradford assay. A total of 50 μg of protein was used for SDS-PAGE at 12% concentration of the resolving gel and run at 80 V for 3 h. The proteins were then transferred onto PVDF membrane, blocked using 5% skimmed milk and subsequently probed using primary and secondary antibodies. The experiment was carried out in two biological replicates and the quantification is shown for one experiment.

# RESULTS

## Systematic identification of RNA editing events in glioma and control brain tissues

The advent of Next Generation Sequencing (NGS) has facilitated identification of novel RNA editing events. The availability of RNA sequencing as well as, matched whole exome sequencing (WES) data provide us with the opportunity to study RNA editing events that are present in the RNA but do not occur in the DNA. Hence, we aimed to unravel RNA editing events in glioma pathogenesis through analysis of RNA-seq and WES data. For this purpose, data of glioma tumor tissue samples from The Cancer Genome Atlas (TCGA) was analyzed. Further, RNA-seq samples from Chinese Glioma Genome Atlas (CGGA), glioma derived cell lines from Cancer Cell Line Encyclopedia (CCLE) and control brain samples from Sequence Read Archive (SRA) were analyzed to find out potential RNA editing events.

A brief schematic of the overall workflow has been given in Fig. 1. Whole RNA sequencing data from TCGA, CCLE, CGGA and SRA were downloaded and aligned to the human reference genome hg19. Post alignment, variants were called in comparison to hg19. These variants were filtered through various datasets—dbSNP, 1,000 genome project and ESP6500 to remove variations that are essentially polymorphisms. Exome sequencing data for glioma samples was available only in TCGA dataset. Hence, the variants were compared with corresponding whole exome sequencing data only in TCGA glioma

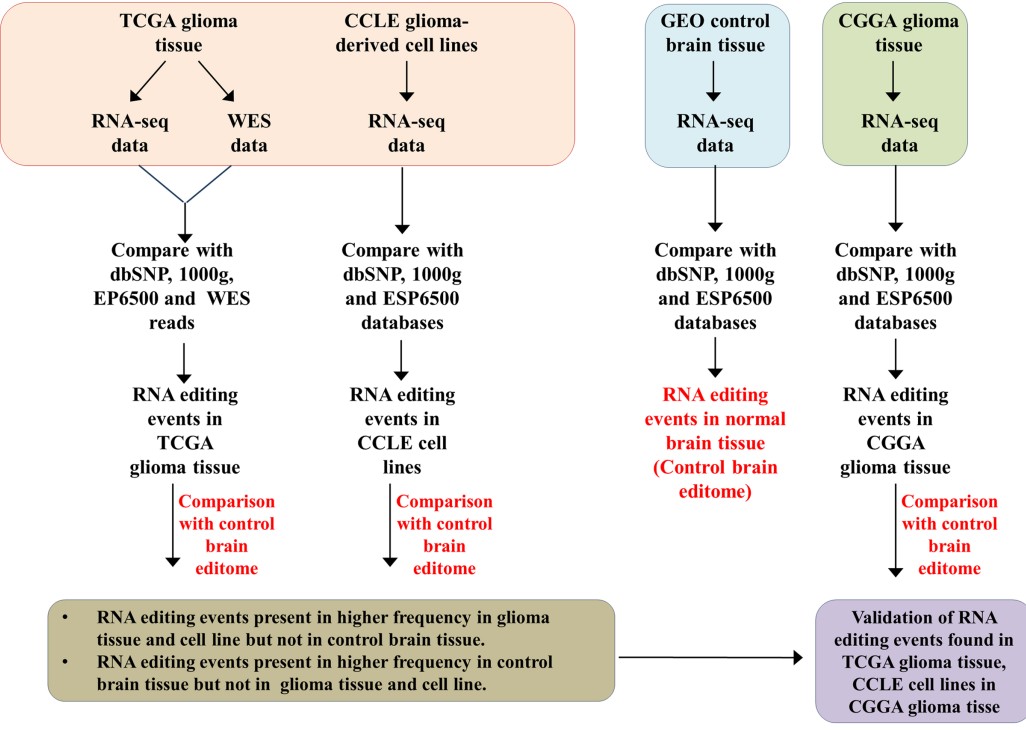

**Figure 1 Scheme of the computational analyses carried out in the study.** The data used for this study were from TCGA glioma samples (*n* = 683), CGGA glioma samples (*n* = 274), CCLE glioma cell lines (*n* = 45) and SRA control brain samples (*n* = 63).

samples to eliminate individual-specific genetic changes. RNA editing events identified from glioma samples were compared with control brain RNA editing events. We validated differential RNA editing events in TCGA glioma compared to control brain samples in the independent CGGA cohort (Fig. 1; Fig. S1).

## The distribution of RNA editing events and editing ratio

We found higher frequency of RNA editing events in Alu repeat region (94.81%) compared to non-Alu repeat (2.24%) and non-repeat (2.95%) regions in control brain samples (Fig. 2A). We observed that majority of RNA editing were found in intronic (70.82%), followed by intergenic (17.91%), non-coding RNAs (6.66%), 5′flank (2.93%), UTR (1.5%) and exonic (0.19%) regions in the control brain samples (Fig. 2B). We found majority of editing events were ADAR specific changes (A-to-I that is, A-to-G or T-to-C) followed by APOBEC specific changes (C-to-U that is, C-to-T or G-to-A) in control brain samples (Fig. 2C). We found A-to-I editing events in higher frequency in the non-coding regions of the genome, such as, UTR, intronic, intergenic, non-coding RNAs and 5′flank regions; whereas we found higher frequency of C-to-U editing only in exonic region of the genome in control brain samples (Fig. 2D). We also identified elevated levels of editing ratio in Alu repeat compared to non-Alu repeat and non-repeat regions in control brain suggesting that apart from number of editing events, editing ratio is also more in Alu repeat region (Fig. 2E). We observed similar RNA editing distribution for

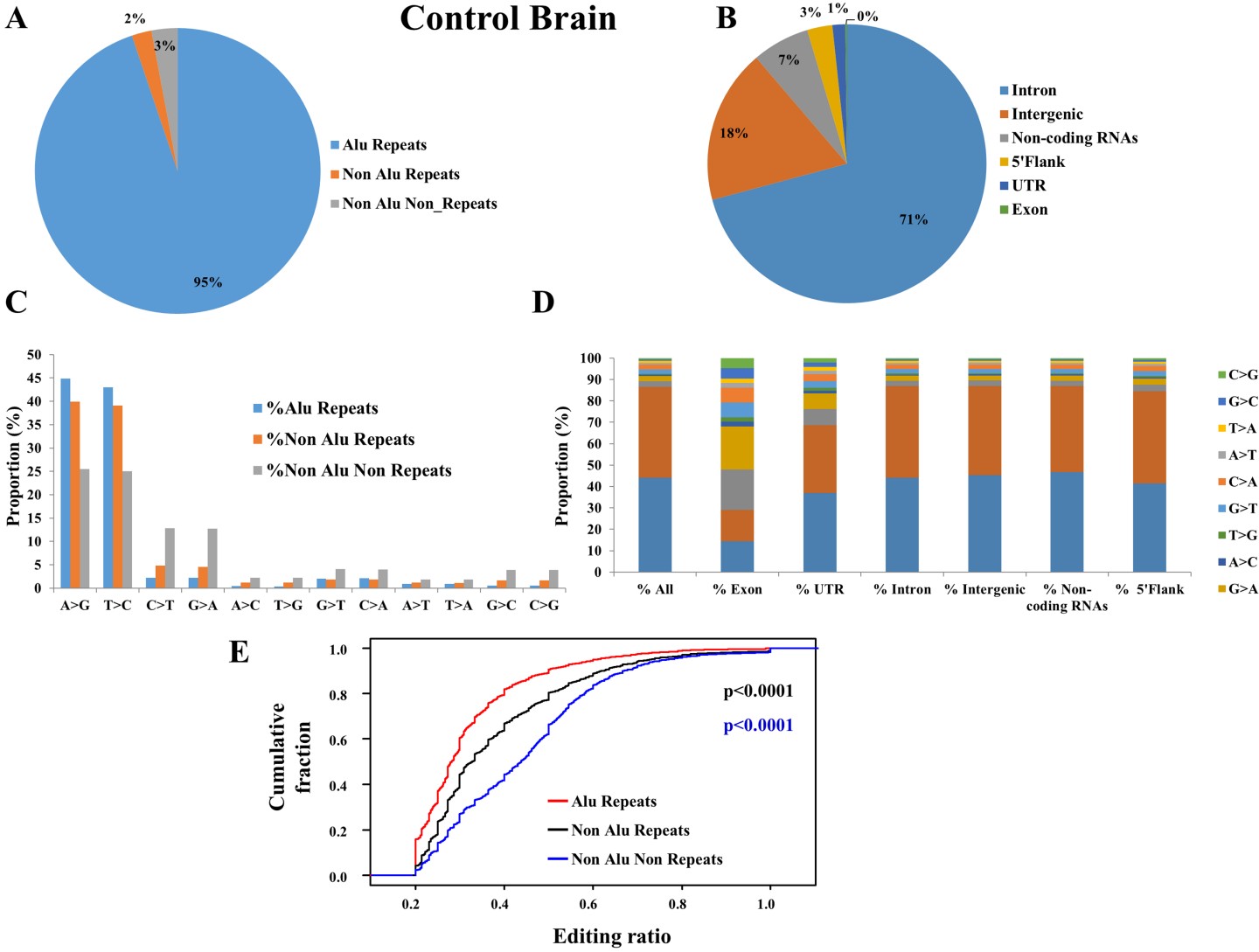

**Figure 2 Distribution of editing events across the genome for control brain samples.** (A) Distribution of total editing events in different regions of the genome: in Alu repeat, non-Alu repeat and non-repeat regions. (B) Distribution of total editing events in different regions of the genome: in exonic, UTR, intergenic and intronic regions. (C) Distribution of different types of editing events in different regions of the genome (Alu repeat, non-Alu repeat and non-repeat). (D) Distribution of types of RNA editing events across different portions of the genome (exonic, intronic, intergenic and UTR). (E) Cumulative distribution function of editing levels of Alu repeat, non-Alu repeat and non-repeat regions for control brain samples. Significance testing was performed using Mann–Whitney *U*-test. Black color *p* value is between alu repeat regions vs. non-alu repeat regions; red color *p* value is between alu repeat regions vs. non-repeat regions.

TCGA LGG (Fig. S2), TCGA GBM (Fig. S3), CGGA GBM (Fig. S4), CGGA LGG (Fig. S5) and CCLE glioma cell lines (Figs. S6 and S7). We observed no significant differences in the global RNA editing distribution patterns between the normal and the glioma samples. From this analysis, we conclude that majority of the editing events were ADAR-specific A to I changes that occur in the non-coding regions of the genome.

## The distribution of RNA editing ratio by glioma phenotype

We identified large number of RNA editing sites in all the data sets. We chose those editing events that were covered by 10 or more reads and defined them as high confidence

RNA editing events. We found higher number of high confidence RNA editing events ($n$ = 734,436) in control brains samples as compared to glioma samples in different data sets such as TCGA LGG ($n$ = 221,049), TCGA GBM ($n$ = 220,954), CGGA LGG (187,757), CGGA GBM (134,509) and CCLE glioma cell lines (140,952) (Fig. S8A; Tables S2–S7). We calculated editing ratio (number of edited reads divided by total number of reads at a given site) for all the samples. A-to-I RNA editing, the most prevalent editing event found in our data, is carried out by the ADAR family of enzymes (*Keegan, Gallo & O'Connell, 2001*; *Nishikura, 2010*; *Wulff, Sakurai & Nishikura, 2011*). We found elevated levels of ADAR in glioma samples as compared to control brain samples, although overall editing was lesser in glioma (Figs. S8A and S8B). Interestingly, we found reduced levels of ADARB1 and ADARB2 in glioma samples, indicating that these two ADAR family enzymes may play an important role in A-to-I RNA editing events in glioma (Figs. S8A, S8C and S8D). We identified that RNA editing ratios were higher in control brain as compared to LGG and GBM in TCGA (Fig. 3A) and CGGA (Fig. 3B) datasets. Similarly, we found RNA editing ratios were higher in control brain as compared to glioma cell lines (Fig. 3C). From this, we can say that overall RNA editing events occur at a reduced frequency in glioma compared to normal and ADAR family enzymes ADARB1 and ADARB2 may be responsible for this decreased editing. Hence, RNA editing play a role in maintaining normal functions of the brain and decrease in editing leads to diseased condition.

## Differential RNA editing events in glioma

We compared RNA editing ratio of control brain samples with LGG samples of TCGA cohort to find out differential RNA editing events in LGGs. We found increased level of editing ratio in 176 editing events in TCGA LGGs as compared to control brain samples and 4,842 editing events had a reduced level of editing ratio in TCGA LGGs as compared to control brain samples (Fig. 4A; Table S8). Similarly, we found increased editing ratio of 185 editing events in CGGA LGGs as compared to control brain samples and 3,287 editing events exhibited reduced editing ratio in CGGA LGGs as compared to control brain samples (Fig. S9A; Table S9). Next, we compared RNA editing ratio of control brain samples with TCGA GBM samples. We identified higher editing ratio of 421 editing events in TCGA GBM as compared to control brain and reduced editing ratio of 3,054 editing events in TCGA GBM as compared to control brain samples (Fig. 4B; Table S10). Similarly, we also observed that 237 editing events were having higher editing ratio in CGGA GBM samples as compared to control brain samples and 3,531 editing events were having reduced editing ratio in CGGA GBM samples as compared to control brain samples (Fig. S9B; Table S11). Next, we compared RNA editing ratio of glioma cell lines with control brain samples. We found 1,171 editing events had higher editing ratio in glioma cell lines as compared to control brain samples and 2,842 editing events had reduced levels of editing in glioma cell lines as compared to control brain samples (Fig. S9C; Table S12). We observed that majority of RNA editing events were reduced in LGGs and GBM samples as compared to control brain samples. Also, majority of significant downregulated editing

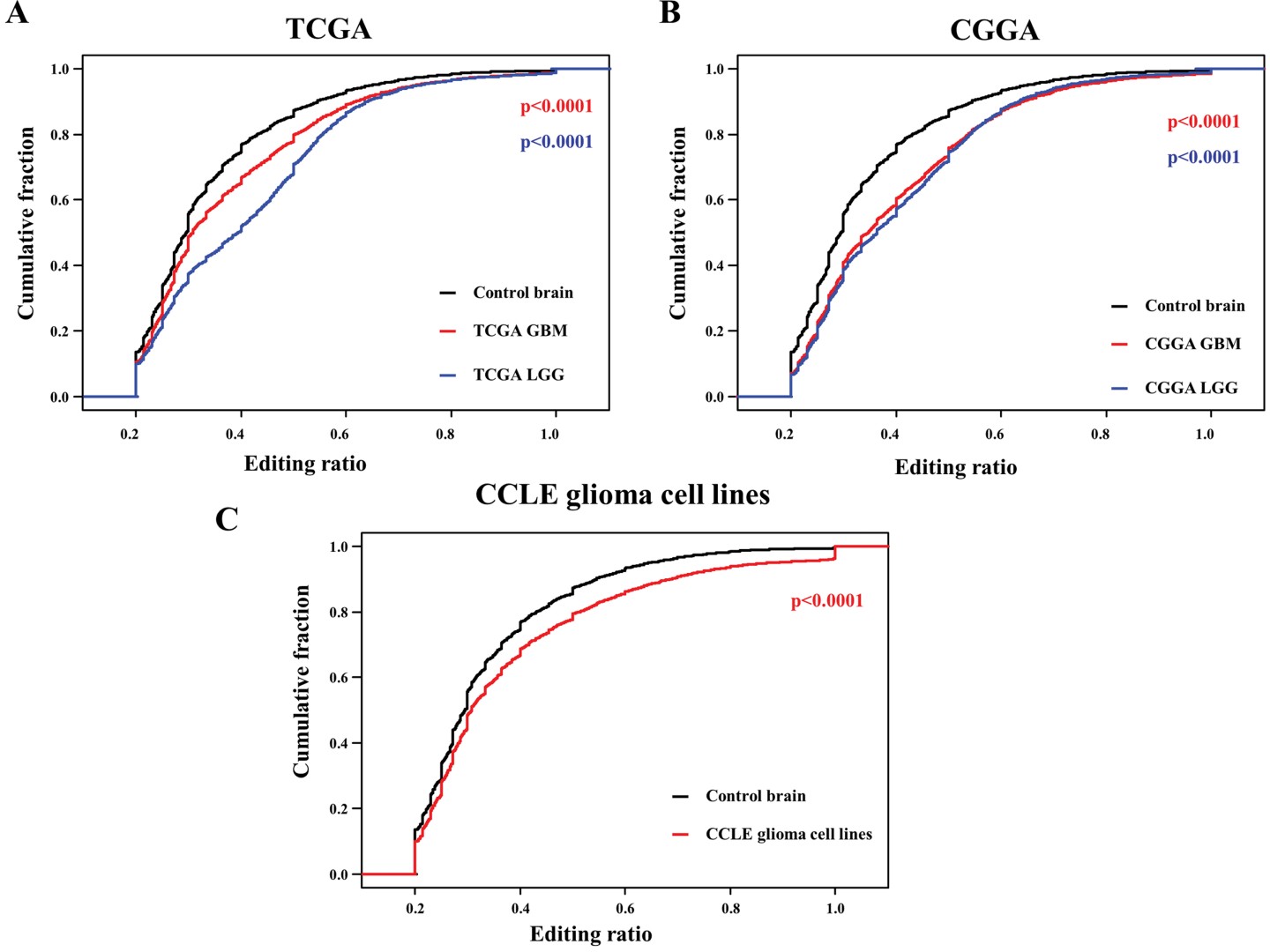

**Figure 3** **The distribution of RNA editing ratio by phenotype.** (A) Cumulative distribution function of editing ratios comparison for control brain ($n = 63$) samples, TCGA GBM ($n = 173$) samples and TCGA LGG ($n = 511$) samples. Red color $p$ value is between control brain samples vs. TCGA GBM samples; blue color $p$ value is between control brain samples vs. TCGA LGG samples. (B) Cumulative distribution function of editing ratios comparison for control brain ($n = 63$) samples, CGGA GBM ($n = 100$) samples and CGGA LGG ($n = 174$) samples. Red color $p$ value is between control brain samples vs. CGGA GBM samples; blue color $p$ value is between control brain samples vs. CGGA LGG samples. (C) Cumulative distribution function of editing ratios comparison for control brain ($n = 63$) samples and CGGA glioma cell lines ($n = 45$). Significance testing was performed using Mann–Whitney $U$-test.                     

events between LGG samples vs. control brain samples and GBM samples vs. control brain samples were overlapping (Figs. 4C and 4D).

Next we checked the concordance of RNA editing events in TCGA and CGGA data sets. We found 85.54% of RNA editing events (both upregulated and downregulated) in CGGA LGGs were present in TCGA LGGs with downregulated editing events showing a higher overlap (Figs. 5A and 5B). Similarly, we found that CGGA GBM RNA editing events had a concordance of 72.11% with TCGA GBM RNA editing events (Figs. 5C and 5D).

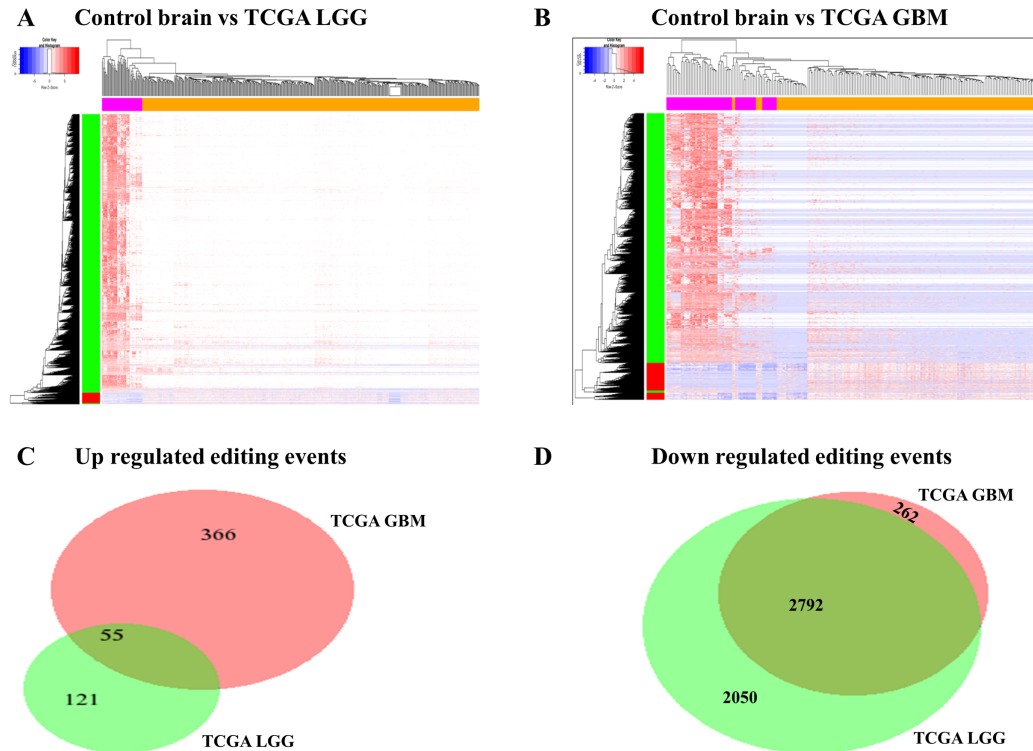

**Figure 4 Differential RNA editing events in glioma.** (A) Heat map representing two-way hierarchical clustering of significant differential RNA editing events in TCGA LGG ($n$ = 511) vs. control brain ($n$ = 63) samples. Samples are shown vertically, magenta color represents control brain and orange color represents TCGA LGG tumors. RNA editing are shown in rows, red color represents upregulated editing events in TCGA LGG tumors as compared to control brain samples and blue color represents down-regulated editing events in TCGA LGG tumors as compared to control brain samples. (B) Heat map representing two-way hierarchical clustering of significant differential RNA editing events in TCGA GBM ($n$ = 172) vs. Control brain samples ($n$ = 63). Samples are shown vertically, magenta color represents control brain and orange color represents TCGA GBM tumors. RNA editing are shown in rows, red color represents upregulated editing events in TCGA GBM tumors as compared to control brain samples and blue color represents downregulated editing events in TCGA GBM tumors as compared to control brain samples. (C) and (D) Venn diagram representing the common number of differentially editing events in control brain samples vs. TCGA LGG tumors and control brain samples vs. TCGA GBM tumors.

Hence, majority of the differential editing events were found to be present in both TCGA and CGGA datasets.

## Pathway analysis of genes exhibiting differential RNA editing in glioma

In depth analysis of the genes that underwent RNA editing identified several interesting facts. Two genes, MLLT6 and MPP2, that were most edited in control brain compared to GBM have been implicated in leukemogenesis and cell adhesion/cytoskeleton regulatory functions (*Rademacher et al., 2016*; *Saha et al., 1995*). We also carried out Gene Ontology (GO) analysis to find pathways that get regulated by editing to understand further the role of RNA editing in gliomagenesis. A total of 71, 50, 56, 44 and 28 GO biological processes were enriched for TCGA LGG, TCGA GBM, CGGA LGG, CGGA GBM and CCLE glioma cell lines respectively (Fig. 6A). We observed 14 pathways to be common

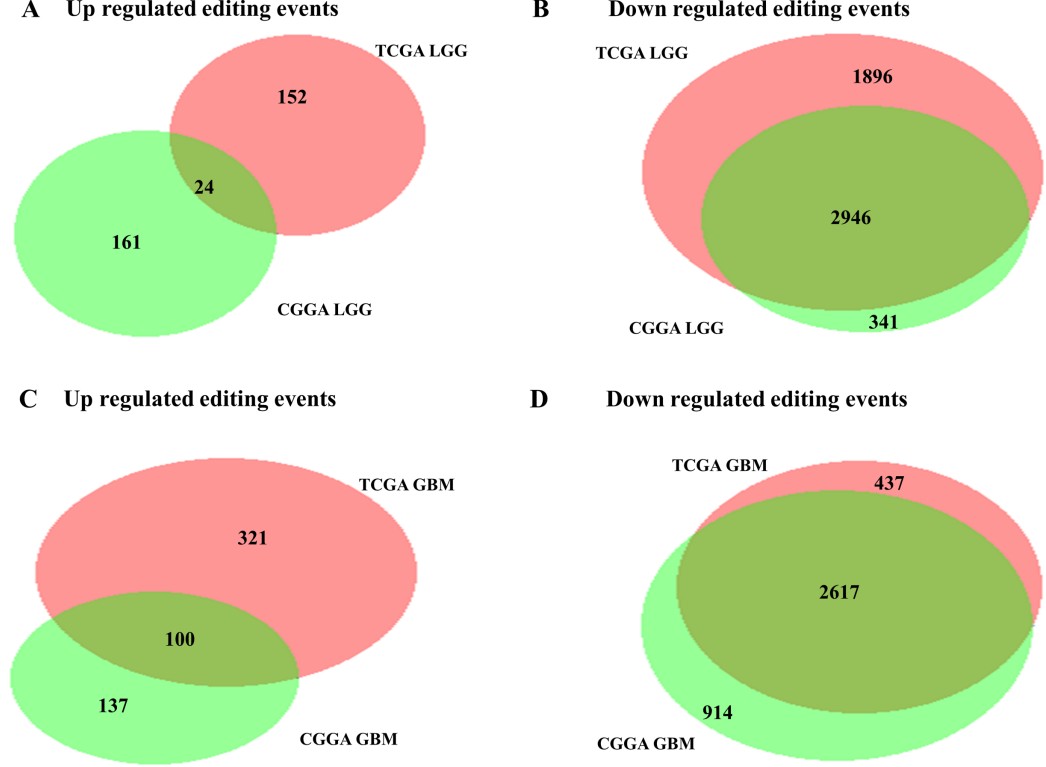

**Figure 5 Validation of differential RNA editing events in independent dataset.** (A) and (B) Venn diagram representing the common number of differentially editing events in TCGA LGG tumors and CGGA LGG tumors. (C) and (D) Venn diagram representing the common number of differentially editing events in TCGA GBM tumors and CGGA GBM tumors.

among all datasets which corresponded to neuronal function and protein translocation (Figs. 6A and 6B). To identify proteins that are importantly regulated by editing in glioma, we identified genes from the above 14 pathways that get commonly altered in all the five datasets (Fig. 6C). A total of 63 genes were found to be altered through editing that belong to the above pathways. Next, we evaluated the distribution of the above 63 edited genes across the genome which revealed only 0.78% of genes to be missense in nature (Fig. 6D). A total of 5 out of the 63 genes that were enriched in the above pathways were found to be carrying missense type editing (Fig. 6E) and these genes include GRIA2, GABRA3, GRIK2, NOVA1 and GRIK1.

## Loss of editing in the coding region of GABRA3 cause aggressiveness of Glioma

It is evident that the majority of the aforementioned missense edited genes, that came up in our pathway enrichment analysis, play roles in neuronal signaling. These genes were found to be highly edited in normal brain and editing ratio and percentage of edited samples decreased significantly in glioma patients. We performed further investigation on GABRA3 because we found it to be highly edited (X: 151358319, A > G; pI342M) in normal brain (73%) with an average editing ratio of 0.8 while the editing percent and ratio

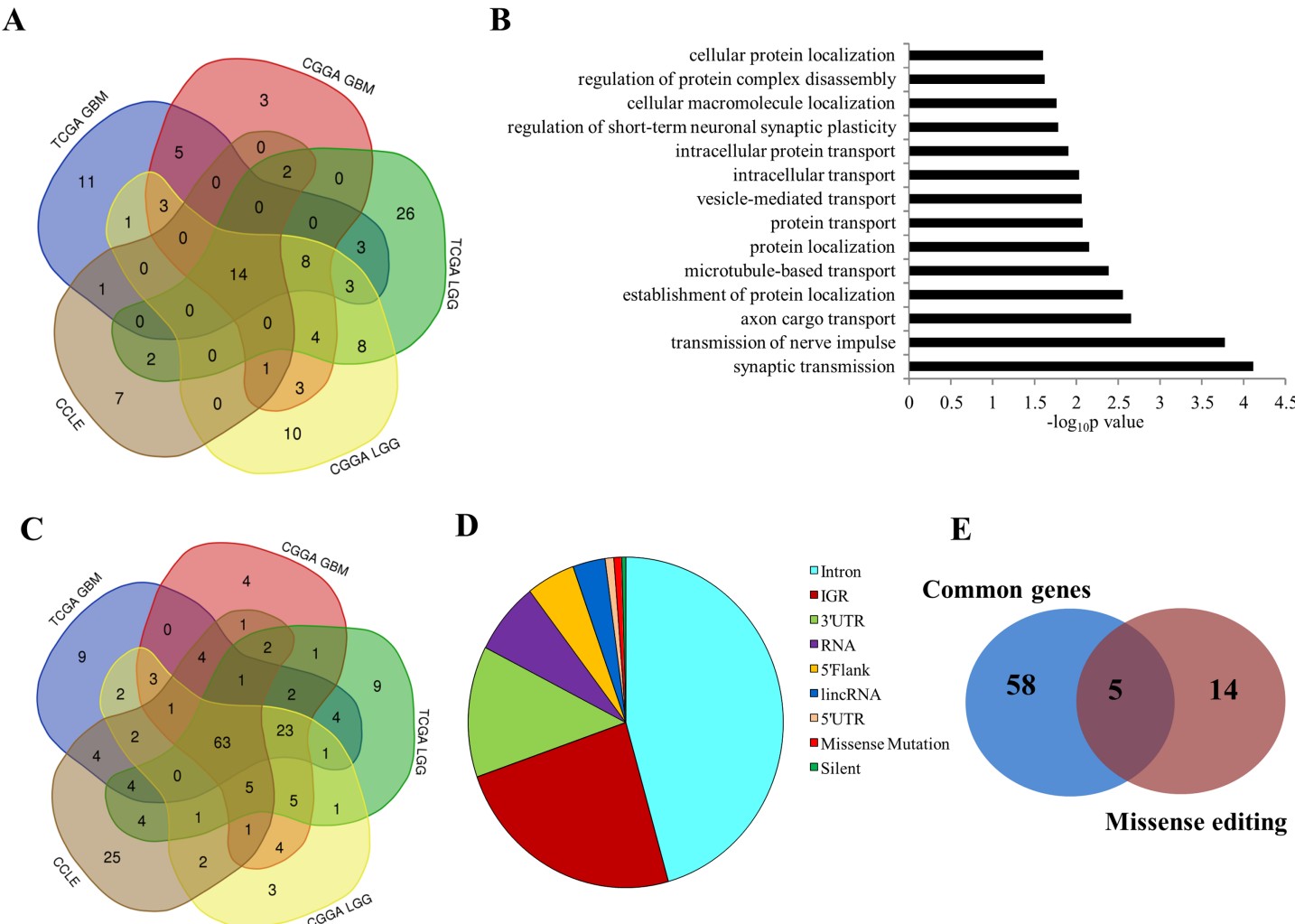

**Figure 6 Pathway analysis of edited events.** (A) Venn diagram represents Gene Ontology (GO) pathways of genes altered by RNA editing in TCGA GBM, CGGA GBM, TCGA LGG, CGGA LGG and CCLE samples. (B) The 14 pathways common between the five datasets—TCGA GBM, CGGA GBM, TCGA LGG, CGGA LGG and CCLE. (C) Venn diagram represents genes from the above 14 pathways that are common between the five datasets. (D) Distribution of the editing events across the different regions of the genome. (E) Venn diagram represents overlap between the missense edited genes and the genes enriched in our pathway analysis that are common among the five datasets.

reduced significantly in GBM samples (5.2% and 0.5 respectively). Indeed, GABRA3 editing reduced significantly in all datasets—TCGA LGG, CGGA LGG, TCGA GBM and CGGA GBM compared to control brain (Fig. 7A). Moreover, the transcript level of GABRA3 was found to be significantly lower in LGG and GBM samples of both datasets (Fig. 7B). In fact, the transcript levels of unedited glioma samples (LGG + GBM) were found to be significantly lower compared to edited glioma samples (Fig. 7C) and the RNA levels correlated significantly with the editing levels of the gene ($R = 0.51$; $p$ value < 0.0001; Fig. 7D). Correlation of the overall editing events in all samples taken together with ADAR, ADARB1 and ADARB2 revealed that GABRA3 editing co-relate significantly with all three enzymes (Fig. 7E). In case of normal brain, GABRA3 editing levels correlated significantly with ADAR and ADARB1 levels. For LGG samples (TCGA and CGGA),

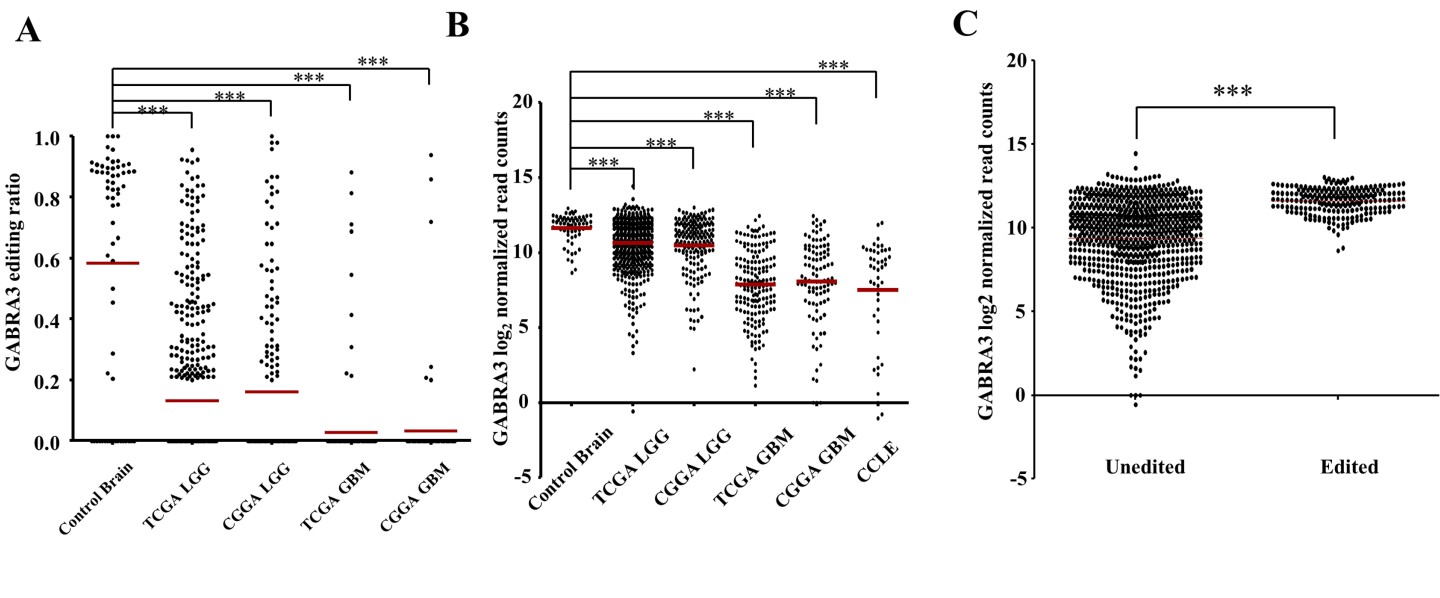

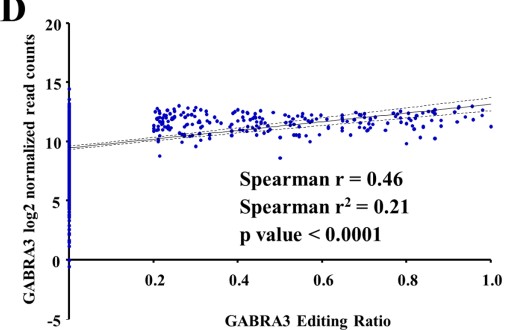

**Figure 7 Regulation of GABRA3 gene by RNA editing event.** (A) Scatter plot representing RNA editing levels of GABRA3 in control brain ($n = 63$) samples, TCGA LGG ($n = 511$), TCGA GBM ($n = 172$), CGGA LGG ($n = 174$), CGGA GBM ($n = 100$) and CCLE glioma cell lines ($n = 45$). (B) Scatter plot representing mRNA expression of GABRA3 in control brain ($n = 63$) samples, TCGA LGG ($n = 511$), TCGA GBM ($n = 172$), CGGA LGG ($n = 174$), CGGA GBM ($n = 100$) and CCLE glioma cell lines ($n = 45$). (C) Scatter plot showing mRNA expression of GABRA3 in GABRA3 edited samples ($n = 217$) vs. GABRA3 unedited samples ($n = 755$). (D) Scatter plot of correlation between RNA editing level of GABRA3 ($n = 972$) and mRNA expression of GABRA3 ($n = 972$). (E) Correlation between mRNA expression of ADAR family of enzymes and RNA editing levels of GABRA3. Significance testing was performed using Mann–Whitney $U$-test. Three asterisks (***) represent a $p$-value of 0.001.

that exhibit moderate levels of editing in GABRA3, it was observed that the editing levels significantly correlate with all three enzymes. However, in GBM samples where GABRA3 editing is very low, the editing ratio did not correlate with any of the ADAR enzymes.

To understand the effect of loss of editing in glioma cells, we exogenously overexpressed edited and unedited forms of GABRA3 (Fig. S10A) in GABRA3 low cell lines LN229 and T98G which harbor no GABRA3 editing (Fig. S10B). Overexpression of edited GABRA3 in LN229 and T98G cells led to reduced migration and invasion potential of glioma cells compared to those expressing the unedited form (Figs. 8A–8F). Further, we predicted effect of amino acid change (I342M) as a result of RNA editing event on GABRA3 protein structure using PolyPhen-2 (*Adzhubei et al., 2010*), SIFT (*Kumar,*

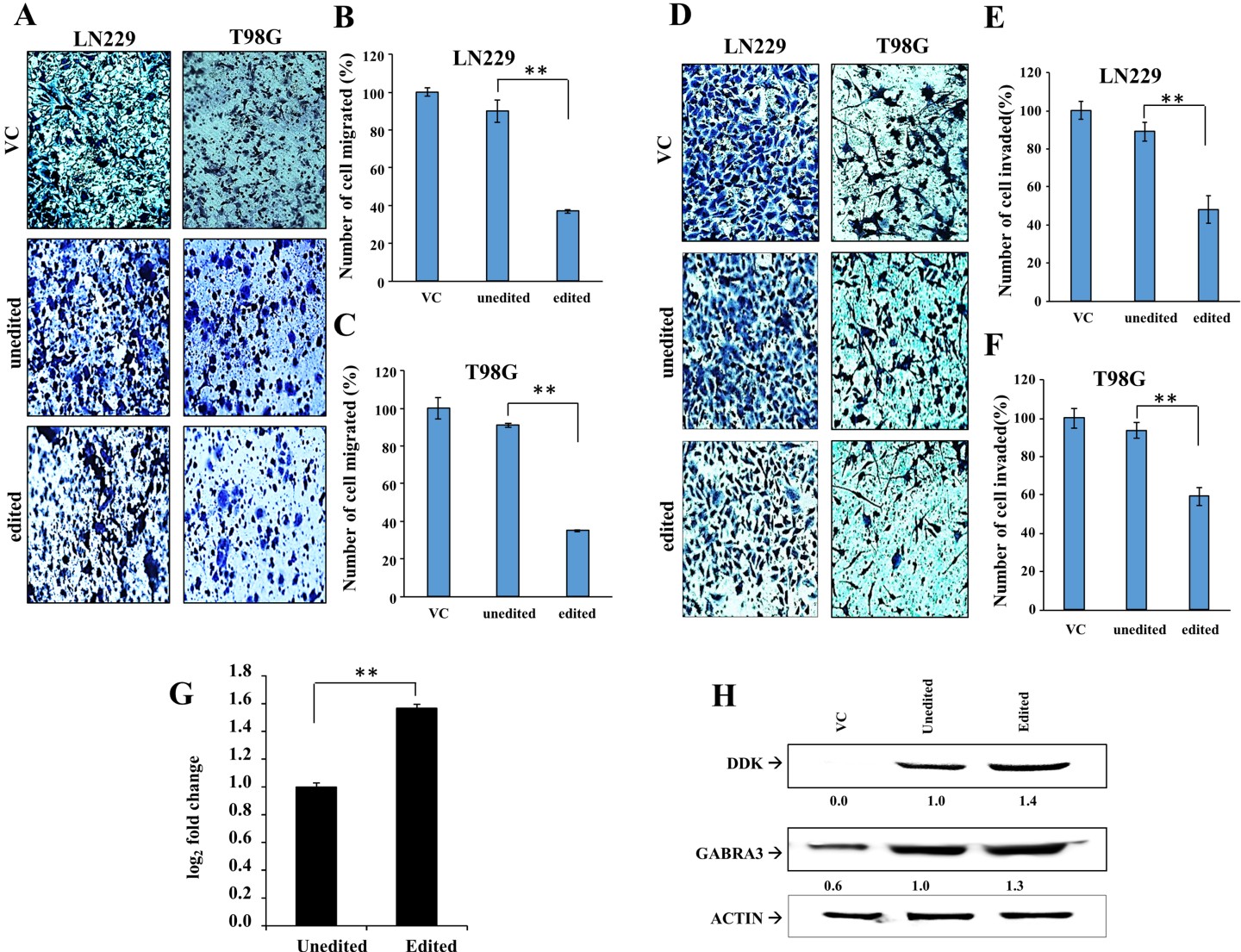

**Figure 8 Effect of GABRA3 editing on glioma cell migration and invasion.** (A) Migration of LN229 and T98G cells overexpressing vector control (VC), edited and unedited GABRA3 gene. (B) and (C) Quantification of migration of LN229 and T98G cells overexpressing overexpressing vector control (VC), edited and unedited GABRA3 gene. (D) Invasion potential of LN229 and T98G cells overexpressing vector control (VC), edited and unedited GABRA3 gene. (E) and (F) Quantification of invasion potential of LN229 and T98G cells overexpressing vector control (VC), edited and unedited GABRA3 gene. Two asterisks (**) represent a *p*-value of < 0.01. Significance testing was performed using Mann–Whitney *U*-test. (G) RNA levels of GABRA3 gene in LN229 glioma cells having edited/unedited GABRA3 overexpressed ectopically. The RNA levels were normalized with respect to vector control (VC). (H) Protein levels of GABRA3 gene in LN229 glioma cells having edited/unedited GABRA3 overexpressed ectopically in comparison to VC. The quantification has been provided underneath each blot with the levels of edited GABRA3 normalized to 1.0.

Henikoff & Ng, 2009), PROVEN (*Choi et al., 2012*) and MutationAssesor (*Reva, Antipin & Sander, 2011*). Since all tools predicted neutral function of RNA editing events, we measured the transcript levels of overexpressed GABRA3 (normalized to endogenous GABRA3). This analysis revealed that loss of editing indeed leads to decreased transcript levels (Fig. 8G; Fig. S10C). Consequently, the protein levels of GABRA3 were found to be reduced in unedited GABRA3 overexpressing cells compared to those expressing the

edited form (Fig. 8H; Figs. S11 and S12). In LN229 and T98G cells, we found 39% and 47% decrease in protein levels of unedited GABRA3 compared to edited GABRA3 overexpressing cells respectively. From this, we conclude that loss in editing of GABRA3 in glioma leads to reduced transcript and protein levels of GABRA3 which results in reduced inhibition of invasion and migration functions thus creating an aggressive GBM.

## DISCUSSION

Next generation sequencing techniques have helped to identify novel RNA editing events in many studies (*Bahn et al., 2012*; *Chan et al., 2014*; *Chen et al., 2013*; *Choudhury et al., 2012*; *Ju et al., 2011*; *Li et al., 2011*; *Ramaswami et al., 2012*, *2013*). In this study, we have analyzed RNA-seq data of 685 lower-grade glioma tumors, 272 GBM tumors, 45 glioma cell lines and 63 control brain samples from publicly available datasets.

A-to-I changes are the most common type of RNA editing events in mammals, especially in humans (*Gott & Emeson, 2000*). A-to-I editing events, catalyzed by ADAR family of enzymes, are most prevalent in the Alu repeat regions because of the double stranded RNA structures formed by inverted Alu repeats that spread across the genome (*Levanon et al., 2004*; *O'Connell et al., 1995*). In this study, we found higher frequencies of RNA editing events in Alu repeat regions compared to non-Alu repeat and non-repeat regions in normal, LGG and GBM samples from all datasets. We also identified increased levels of editing ratio in Alu repeat compared to non-Alu repeat and non-repeat regions in all datasets suggesting that apart from the number of editing events, editing ratio is also more in Alu repeat region. We also observed that majority of editing events are ADAR specific changes (A-to-I that is, A-to-G or T-to-C) followed by APOBEC specific changes (C-to-U that is, C-to-T or G-to-A) similar to previously reported studies (*Ramaswami et al., 2012*, *2013*). A-to-I RNA editing events were found to be higher in the non-coding regions of the genome. This could be because the ADAR family of enzymes that bring about the A-to-I editing recognizes Alu repeat regions for binding and these repeat sequences are sparsely present in the coding regions of the genome (*Grover et al., 2003*). Conversely, we also observed that majority of the editing in the exonic region are of the C-to-U type catalyzed by the APOBEC family of enzymes (*Rosenberg et al., 2011*).

Our study revealed RNA editing levels to be higher in control brain samples and it is significantly reduced in glioma samples which correlate with the RNA expression levels of the ADAR enzymes ADARB1 and ADARB2. Indeed, it was shown in a previous report that ADAR enzymes are downregulated in glioma and overexpression of the gene leads to tumor suppressive effects (*Maas et al., 2001*). However, out of the three ADAR family enzymes, ADAR was found to be expressed in glioma samples which could be the primary source of A-to-I editing in glioma. RNA editing events which are downregulated in glioma compared to control brain samples showed a significantly higher concordance between different datasets. This could be because in general, ADAR enzymes are downregulated in glioma leading to reduced editing while the upregulated editing events may be spurious.

The editing events of GRIA2 and GABRA3 as unearthed by this study have been previously reported to be essential for the functioning of normal mammalian brain (*Ohlson et al., 2007*; *Sommer et al., 1991*). Moreover, GRIA2 underediting was found to be

responsible for aggressive phenotype in GBM (*Oakes et al., 2017*). RNA editing in GABRA3 was found to be present in the coding region of the gene which led to Isoleucine to Methionine conversion in the 342th amino acid position of the protein. ADAR and ADARB1 are the major ADAR enzymes in the brain. Hence, we observed a significant correlation between the RNA levels of GABRA3 with these two enzymes in the control brain and this correlation was lost in GBM samples where global editing levels decrease drastically. We further studied the effect of this missense RNA editing in GABRA3.

From literature, we see that GABRA3 editing increases during brain development in rats with a simultaneous decrease in the expression (*Daniel et al., 2011*). However, in our study, we reveal that loss of editing in GABRA3 led to reduced gene expression levels in LGG and GBM samples from both TCGA and CGGA datasets. Moreover, experimental studies revealed that loss of editing leads to reduced levels of GABRA3 RNA and protein accompanied by a more migratory and invasive phenotype of the glioma cell lines. Although we test experimentally in a small sample set ($n = 2$), the results are confirmed by similar regulation in the larger TCGA and CGGA datasets.

This opposing regulation of GABRA3 by editing could be cell type-specific and development-specific. In fact, it is possible that the regulation in adult brain is different. In the healthy adult human brain, editing of GABRA3 is found in different regions of the brain irrespective of enrichment of neuronal or glial populations (*Azevedo et al., 2009*). Moreover, the editing of GABRA3 does not alter even with age (*Holmes et al., 2013*), suggesting the importance of the editing in non-cancerous brain tissue. But during early developmental stages, the effect of GABRA3 editing is probably specific to the neuronal population due to the requirement of a switch of GABA response from excitatory to inhibitory post-synaptic potentials (Ben-Ari et al., Nat. Rev. Neur., 2002).

In a recent study, it is shown that loss of editing in GABRA3 leads to invasive phenotype in breast cancer (*Gumireddy et al., 2016*). In this study, we reveal the tumor suppressive nature of editing of GABRA3 in glioma which stresses on the requirement for the gene to be edited in normal scenario. Moreover, loss of editing of GABRA3 in GBM was accompanied by a decrease in RNA levels as seen in the patient tissue samples. In case of the patient data, we compared with control brain tissue which comprises of both neuronal and glial cells. In this study, we could not specifically evaluate the contribution of GABRA3 editing from glial cells alone. This is a limitation of the present study. However, from experimental data in GBM cell lines, we confirmed that loss of editing indeed led to a decrease of both RNA and protein level of GABRA3. Thus, GABRA3 editing in glioma may lead to decrease in the protein which might be responsible for the aggressive tumor phenotype but this requires further validation in future.

## CONCLUSION

Thus, the present study unfolds the entire RNA editome landscape of glioma tissues which will help scientists in understanding the importance of post-transcriptional sequence alterations in diseased conditions. Furthermore, the effect of loss of editing of GABRA3 during glioma progression highlights the importance of RNA editing for the maintenance

of tissue homeostasis. The study will open up novel avenues of research and therapeutic interventions for glioma.

## ACKNOWLEDGEMENTS

The results published here are in whole or part based upon data generated by The Cancer Genome Atlas pilot project established by the NCI and NHGRI. Information about TCGA and the investigators and institutions that constitute the TCGA research network.

### Funding

Kumaravel Somasundaram received a research grant from DST (SUPRA SEED), CSIR, ICMR and DBT, Government of India and is a J.C. Bose Fellow of the Department of Science and Technology. The Microbiology and Cell Biology Department received funding from DST-FIST, DBT grant-in-aid and UGC (Centre for Advanced Studies in Molecular Microbiology) for infrastructure. The funders had no role in study design, data collection and analysis, decision to publish, or preparation of the manuscript.

### Grant Disclosures

The following grant information was disclosed by the authors:
DST (SUPRA SEED), CSIR, ICMR and DBT, Government of India.
J.C. Bose Fellow.
DST-FIST, DST (SUPRA SEED), IISc-DBT partnership program, ICMR and UGC.

### Competing Interests

Kumaravel Somasundaram is an Academic Editor for PeerJ.

### Author Contributions

- Vikas Patil conceived and designed the experiments, performed the experiments, analyzed the data, prepared figures and/or tables, authored or reviewed drafts of the paper, and approved the final draft.
- Jagriti Pal conceived and designed the experiments, performed the experiments, analyzed the data, prepared figures and/or tables, authored or reviewed drafts of the paper, and approved the final draft.
- Kulandaivelu Mahalingam conceived and designed the experiments, authored or reviewed drafts of the paper, and approved the final draft.
- Kumaravel Somasundaram conceived and designed the experiments, prepared figures and/or tables, authored or reviewed drafts of the paper, and approved the final draft.

### Data Availability

The 174 LGG and 100 GBM RNA sequencing samples are available from Chinese Genome Glioma Atlas (CGGA) via SRA (SRP027383) (*Bao et al., 2014*). The RNA

sequencing data of control brain samples (SRP033725, SRP045638, SRP044668) ($n = 63$) are available from SRA (*Akula et al., 2014*; *Gill et al., 2014*; *Jaffe et al., 2015*).

## Supplemental Information

Supplemental information for this article can be found online at http://dx.doi.org/10.7717/peerj.9755#supplemental-information.

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
