# Peer review of "Global RNA editome landscape discovers reduced RNA editing in glioma: loss of editing of gamma-amino butyric acid receptor alpha subunit 3 (GABRA3) favors glioma migration and invasion"

_PeerJ, doi:10.7717/peerj.9755_

## Round 0.1 · original submission · Major Revisions

Dear Prof. Kumaravel,

Based on the advice received, I feel that your manuscript could be reconsidered for publication should you be prepared to incorporate major revisions. When preparing your revised manuscript, you are asked to carefully consider the reviewer comments and submit a list of responses to the comments. You are kindly requested to also check the manuscript for spelling mistakes and grammatical errors.

Reviewer 1 ·

Basic reporting

The manuscript from Vikas Patil et al. reports the loss of editing of gamma-amino butyric acid receptor alpha subunit 3 (GABRA3) favors glioma migration and invasion using global RNA editome landscape. Specifically, in this study, the authors carried out a comprehensive analysis of RNA sequencing data of normal brain, lower grade Glioma (LGG) and glioblastoma (GBM) samples from publicly available datasets. They provided evidence to understand the distribution of RNA editing events across different regions of the genome, the comparison of editing events between normal, LGG and GBM samples, the regulation of ADAR enzymes in gliomas and the differential editing events between the different types of gliomas. Among them, a total of 5 out of the 63 pathway-related genes, including GRIA2, GABRA3, GRIK2, NOVA1 and GRIK1, were found to be carrying missense type editing from their analysis. Hence, they performed some experiments confirming these findings.

Overall, it would be good for the authors to revise the manuscript by a native speaker.

The bioinformatics analyses are quite thorough and complete, and overall this reads like a very high-quality RNA sequencing report. However, additional data should be performed in the experimental section for the validation of bioinformatics works. As the materials are available in this research, I would recommend performing the additional analysis of RNA and protein levels of GABRA3 gene in T98G cell line as well. Moreover, it would be good to see how GRIA2 function, as it is one of interesting finding in this research that has been previously reported to be essential for the functioning of the normal mammalian brain as the authors stated.

Experimental design

The transfection of DNA plasmids into cell lines for the overexpressing experiments should be obviously described. For reference and repeatable in other studies, the concentration of plasmid DNA and experimental condition should be stated.

The cultural condition for RNA extraction should be described in the materials and method section. Also, please add the information of the primers designed for Real-Time PCR using in this research.

Please add the description for protein experimental work in this research.

Figure 8: Please provide the figure for cell staining of vector transfection as well.

Please obviously state in the manuscript of how many replications has done throughout laboratory experiments.

Validity of the findings

The data and RNA sequencing seem sound, I would strongly suggest performing additional experiments though as mentioned above.

Additional comments

There are errors typing throughout the manuscript. Please carefully double check in revision version.

Reviewer 2 ·

Basic reporting

see below

Experimental design

see below

Validity of the findings

see below

Additional comments

The paper “Global RNA editome landscape discovers reduced RNA editing in glioma “reports interesting work on abnormal RNA editing in human glioma.
They used sequences from 5 independent datasets for their studies. This included two human glioma databases (TCGA and CGGA) taken from independent patient cohorts, each of which included a cohort of lower grade glioma tissue, and a second cohort to GBM tissue. Finally, the results were validated with a fifth “cohort” of sequence from a cadre of glioma cell lines. After filtering for likely SNPs, sequence from each of the five databases were independently compared to sequences from normal human brain to find differentially edited sequences. The results were robust, in that 72-85% of all edited events were concordant between the TCGA and CGGA datasets.
The authors perform an extensive characterization of editing events, include introns/exons, alu-repeats vs non-alu repeats, etc. Personally, I found this portion a little overwrought with 2 figures and 5 supplementary figures. However, this is perhaps a matter of taste since there are not journal -specific restrictions of figure count
The authors then analyzed abnormally edited sequences from glioma tissue and found that reduced editing was much more frequent than increased editing. Furthermore, they showed that the vast majority of glioma-specific hypo-edited sequences were shared between low grade and high-grade gliomas.
critique about figure 4: My version of figure 4 seems to be a very low-resolution screen shot from the authors computer. It is impossible to see any of the details when zooming in on the figure. This should be replaced by a higher quality version.
They then perform a gene ontology analysis, which identified 14 pathways common to all 5 datasets. Further analysis found a total of 63 abnormally edited transcripts common to all datasets, but only five of these sequences included a missense change in the codon portion of the transcript. Interestingly, all but one of these genes is involved in glutamate/GABAergic neurotransmission.
The work then finishes up with a more in-depth analysis of GABRA3 (GABA Receptor α3 subunit). As previously reported (Ohlson et al., 2007; Rula et al., 2008) most α3 in adult brain is fully edited.
Critique -GABRA3 is expressed in neurons, but it is unclear to me how much is expressed in glia. It would be good for the authors to test whether some of these differences could be confounded by a high proportion of neurons in the control tissues. If this is impossible in this setting, they should at least discuss this as a potential pitfall, and how it could skew their results.
Critique- Figure 7 is confusing. Figure 7A shows the medians as being below the point scatter. If this is because of many sequences with 0% editing, this should be made more clear. Similarly, regression line in 7D has a positive slope, but the line is clearly flat by eye. It seems that this line includes all the transcripts with an editing ratio of “0”, which have a very wide scatter. If so then the regression is based on widely scattered points at “0” which have a slightly lower median GABRA3 expression at the left end of the line, and the invariant GABRA3 expression on the right (editing ratio = 0.2 – 1.0).
They then try to test the significance of these findings by transfecting 2 GBM cell lines with low endogenous GABRA3 expression. One set is transfected with an unedited version, the other is transfected with a version to recapitulate the edited version.
Critique- it seems the authors simply used the relevant genomic sequence to make their ‘unedited” version. However, there is nothing to prevent editing once in the cell. In figure 8E, they report overall GABRA3 mRNA expression, but not whether the mRNA from “unedited” cells expressed a significant amount of edited GABRA3 mRNA. If I am mistaken, and the authors modified the sequence to be “uneditable” please state this in the paper. If not please either measure % editing in those cells, or at least discuss the possible confound.
They also report that the edited version seems to express a higher amount of protein, but it seems to be based on a sample size of “2”. This is not adequate, especially given previous work showed that gabra3 editing actually reduces protein expression, which may contribute to the developmental down regulation of this protein in perinatal brain (Daniel et al., 2011)
Finally, the authors show that GABRA3 expression/editing may be an important determinant of GBM lethality, by measuring tumor migration and invasion rates. This could be the most interesting and compelling part of this entire project. However, technical details about how the assay is performed and quantified are sorely lacking. Indeed, it is unclear whether the data in Figure 8A-D were derived from a single round of transfections, or multiple experiments. This section requires much more explanation of the technique and explanation of the results.

More general critiques
The extremely poor English and ubiquitous typographical errors make the paper essentially unacceptable (in my opinion) at this point without extensive edits. For example, the section titled “editing ratio calculation” is riddled with missing words and improper verb usage. Moreover, multiple instances of typographical errors, such as “GBARa3” and “uneditided” are found throughout the manuscript, figures and tables.
It is unclear if the authors are familiar with the previous literature about GABA signaling in brain tumor. Personally, I think the discussion would be improved by including mention some of previous work on the effects of gabra3 editing on protein expression/function in recombinant systems, as well as overall GABAergic signaling in gliomas.

Annotated reviews are not available for download in order to protect the identity of reviewers who chose to remain anonymous.

Reviewer 3 ·

Basic reporting

The manuscript “Global RNA editome landscape discovers reduced RNA editing in glioma: Loss of editing of Gamma-Amino Butyric acid Receptor Alpha subunit 3 (GABRA3) favors glioma migration and invasion” quantifies and describes the RNA editome in various types of gliomas.

Overall, the manuscript conforms to the structure recommended by the journal viz. Introduction, Materials and Methods, Results, Discussion, Conclusions. Barring few typos, the overall English comprehension and grammar of the article is good and attains standards of a scientific article. However, the figures are poorly presented and need considerable work to make them of publication quality. I also found the method section needs to improve by adding more details and explanation for each procedure. Further, I would appreciate if the authors could provide scripts used for the analysis in order to be transparent as per journals motto.

Below are specific points:

1. Please maintain consistency in the text. For e.g. GABRA3 is written a few places and sometimes GBARA3 is mentioned in the text. Please correct wherever applicable.
2. Mention the full-form at the first occurrence of an abbreviation. Example, GBM in the background section of the abstract
3. Include a proper citation to the datasets used for the study. It would be very helpful for the readers following up on your study.
4. Although the introduction section does a good job of providing the relevant background of the key phenomenon and methods, it lacks proper motivation behind the study. Also, it does not provide any rationale to study the RNA editing of GABRA3 gene. This section can be improved by including the knowledge gap underlying RNA editing and gliomas. Also, refer to specific studies looking at GABRA3 RNA editing. Such information is important and will help to understand the relevance of the present study. Further, the reference to the historical development of RNA editing technologies is unnecessary instead a proper motivation and original research question of the study can be highlighted.
1. In the abstract, “median survival of grade IV glioma” should be rephrased to something like “median survival of patients with…”
2. Please mention any accession number for the 45 Glioma cell lines downloaded from the CCLE-cgHub. Publicly available datasets are constantly been updated. For future studies, it would be important to refer back to the exact samples used in this study. Similarly, mention the GEO accession IDs for data downloaded from Chinese Genome Glioma Atlas.
3. Please cite the tools used mentioned in the RNA editing pipeline, for example, BWA-aligner, Picard, GATK profiler etc.
4. Introduce 1000g and ESP6500 datasets with proper reference.
5. The figures in general needs improvement. For many figures, the font size is very small and the font type is inconsistent. Increase the font size and maintain the same font type for all figures.

Experimental design

1. Please provide a justification for using editing ratio = 0.2 (line 124) as a cutoff for classifying an RNA editing event. Presently it sounds very arbitrary. One way to decide the cutoff would be to use the distribution of all editing ratios.
2. The approach explained in the subsection Differential RNA editing of the Materials and Methods section is not clear. It is unclear the utility of calculating “editing difference”. It is also unclear what was the input to the Mann-Whitney Wilcoxon test? A standard way to refer to this test is to either use Wilcoxon test or Mann-Whitney U-test.
3. The criteria for deciding differential RNA editing - “absolute difference between average RNA editing of tumor samples versus control brain samples is greater than 0.2” is arbitrary. Please provide justification.
4. Another way of deciding differential RNA editing is by using Ficher’s exact test as used in the paper - https://www.pnas.org/content/116/6/2318#sec-10
5. Line 204, There is no C-to-U data shown in figure 2D. If you meant C-to-G edits, please correct accordingly in the main text.
6. Line 214, Please justify why 10 number of reads is defined as high confidence. A better way would be to plot a distribution of reads mapped to the editing events and choose 90th or 95th percentile as the cutoff.
7. Please explain the procedure used to calculate the p-value for the figure 2E and similar supplementary figures.
8. Also, for figure 2E and similar supplementary figures, it would be important to see the probability density function to see the distribution of editing ratio values.
9. In figure 4A and 4B, the x and y-axis are not labeled. Please replace with a high-quality version of these two figures. Include the color legend on the figure apart from the text. The color bar legend is also unreadable. I couldn’t judge the results mentioned in this graph as there is no information.
10. In figure 6B, the x-axis is missing, therefore, the results cannot be interpreted.
11. The term “pathway-related” is ambiguous. Replace with a suitable alternative.
12. In figure 8F, ¬¬¬quantify the amount of protein on the gel. Even though there is a mention of 39% increase there is no quantification provided.
13. Line 296, typo “tis”
14. Figure 7D, there is a discrepancy between the R-squared value shown on the graph and mentioned in the main text (line 283). Please replace with the correct value. Also, it should be R-square and not just “R” (line 283).
15. For the regression shown in figure 7D, explain which method was applied to calculate the p-value.
16. In figure 7A and 7B, mention legend for the horizontal red bar. It does not look like the median value.
17. For figure 7A, B, C, what was the procedure used to calculate the p-value.
18. Line 296, mention citation for PolyPhen-2, SIFT, PROVEN and MutationAssesor.
19. Please mention what are the potential candidates that were highly edited in the normal brain compared to gliomas. It would be important for future studies.
20. The discussion section lacks to highlight the additional knowledge gained by performing this study. For example, it is a well-known fact that there is a high frequency of RNA edits in Alu repeats and the same is found in the study. Try to highlight the relevance of this observation in the context of gliomas.
21. In figure 6E, it is unclear how were genes with missense editing were obtained. Please mention the source or method used to obtain them.

Validity of the findings

1. The data set used in the study should be better described (see Basic Reporting section)
2. Mention statistical test for each figure
3. Link to the data set should be included. In which repository sequencing datasets are made publicly available (e.g. SRA)?
4. For further comments refer section Experimental design

---

## Round 0.2 · Minor Revisions

Dear Prof Somasundaram,

Though the current manuscript has improved drastically from the previous version, some of the points raised by Reviewer 2 were not adequately answered. Please address these queries and resubmit the manuscript.

Reviewer 1 ·

Basic reporting

In this revised manuscript, the authors have addressed all my concerns raised in the previous review report, therefore this version is ready for publication.

Experimental design

All good

Validity of the findings

All good

Reviewer 2 ·

Basic reporting

see attached critique in PDF format

Experimental design

see attached critique in PDF format

Validity of the findings

see attached critique in PDF format

Annotated reviews are not available for download in order to protect the identity of reviewers who chose to remain anonymous.

---

## Round 0.3 · accepted · Accept

The manuscript has improved after the revision can now can be accepted for publication.